# Extend Plastron Longevity on Superhydrophobic Surface Using Gas Soluble and Gas Permeable Polydimethylsiloxane (PDMS)

**DOI:** 10.3390/biomimetics10010045

**Published:** 2025-01-13

**Authors:** Ankit Gupta, Hangjian Ling

**Affiliations:** Department of Mechanical Engineering, University of Massachusetts Dartmouth, Dartmouth, MA 02747, USA; agupta2@umassd.edu

**Keywords:** plastron longevity, gas diffusion, polydimethylsiloxane (PDMS) surface, superhydrophobic surface, plastron recovery, wetting transition

## Abstract

The gas (or plastron) trapped between micro/nano-scale surface textures, such as that on superhydrophobic surfaces, is crucial for many engineering applications, including drag reduction, heat and mass transfer enhancement, anti-biofouling, anti-icing, and self-cleaning. However, the longevity of the plastron is significantly affected by gas diffusion, a process where gas molecules slowly diffuse into the ambient liquid. In this work, we demonstrated that plastron longevity could be extended using a gas-soluble and gas-permeable polydimethylsiloxane (PDMS) surface. We performed experiments for PDMS surfaces consisting of micro-posts and micro-holes. We measured the plastron longevity in undersaturated liquids by an optical method. Our results showed that the plastron longevity increased with increasing the thickness of the PDMS surface, suggesting that gas initially dissolved between polymer chains was transferred to the liquid, delaying the wetting transition. Numerical simulations confirmed that a thicker PDMS material released more gas across the PDMS–liquid interface, resulting in a higher gas concentration near the plastron. Furthermore, we found that plastron longevity increased with increasing pressure differences across the PDMS material, indicating that the plastron was replenished by the gas injected through the PDMS. With increasing pressure, the mass flux caused by gas injection surpassed the mass flux caused by the diffusion of gas from plastron to liquid. Overall, our results provide new solutions for extending plastron longevity and will have significant impacts on engineering applications where a stable plastron is desired.

## 1. Introduction

Recently, a superhydrophobic surface with a large water contact angle and low contact angle hysteresis has been fabricated by mimicking the lotus leaf. The superhydrophobic surface is typically created by combining micro/nano-scale surface roughness with hydrophobic surface chemistry. This surface has a wide range of engineering applications, including reducing hydrodynamic friction drag [1], enhancing heat and mass transfer [2], and protecting engineering surfaces from biofouling [3,4], icing [5], and corrosion [6].

However, one of the main challenges that limit the broad application of superhydrophobic surfaces is the gas diffusion issue [7]. When submerged in a liquid, the superhydrophobic surface traps a thin layer of gas (or plastron) between the surface textures, forming the so-called Cassie–Baxter state [8]. Many of the desired properties of superhydrophobic surface depend on the presence of plastron [9,10]. For example, the plastron supports an effective slip boundary [11,12], which results in friction drag reduction. The plastron is also key for the reduction in bacterial adhesion [3,13]. However, when the liquid is undersaturated with gas, the beneficial gas can be dissolved into the ambient liquid [14,15,16,17,18,19,20,21,22,23,24,25,26,27,28,29,30], leading to the so-called Wenzel state [31]. The transition from the Cassie–Baxter state to the Wenzel state is known as wetting transition [32]. In addition to gas diffusion, other factors, such as turbulent flows [33,34,35,36,37] and pressure [38,39], could also trigger the wetting transition. Currently, understanding and extending the plastron longevity is crucial for the broad application of superhydrophobic surfaces.

In the past two decades, various techniques have been developed to extend the longevity of plastrons on the superhydrophobic surface. For example, Lee and Kim [40] and Lloyd et al. [41] developed a technique to restore the gas layer based on the electrolysis of water. Lee and Kim used an Au-coated superhydrophobic surface as the electrode and applied a voltage between the liquid and the surface to induce water splitting. Similarly, Lee and Yong [42] restored the plastron through solar water splitting. Panchanathan et al. [43] recovered the plastron by using the decomposition reaction of hydrogen peroxide on a superhydrophobic surface prepared with a catalytic coating. Vakarelski et al. [44] and Saranadhi et al. [45] sustained a stable Leidenfrost vapor layer on superhydrophobic surfaces by heating. Several authors [46,47,48] restored the plastron by injecting and spreading a gas bubble on a superhydrophobic surface. A few researchers [49,50,51] also extended the plastron longevity by controlling the dissolved gas concentration in ambient liquid. Finally, fabricating superhydrophobic surfaces on porous materials and injecting gas through the porous surface was also frequently used to sustain the plastron [52,53,54,55,56,57].

The goal of this study is to examine a new method for extending plastron longevity by taking advantage of the gas-soluble and gas-permeable properties of polydimethylsiloxane (PDMS). Many different approaches have been developed to manufacture superhydrophobic PDMS surfaces, including coating PDMS with a highly fluorinated monolayer [58], plasma etching [59,60,61,62], laser texturing [63,64,65], and soft-lithography [66,67]. Extending the plastron longevity on superhydrophobic PDMS surfaces is significant since PDMS is widely applied in tissue engineering and microfluidic devices due to its biocompatibility, thermal stability, nontoxicity, and flexibility [68]. However, the plastron longevity on the superhydrophobic PDMS surface has received less attention. Previous studies have mostly focused on the plastron longevity on a non-gas soluble and non-gas permeable surface [14,15,16,17,18]. Although the gas soluble and gas permeable properties of the PDMS were well documented in the literature [69,70,71], their impact on the plastron longevity has not been well investigated [53]. A few studies have shown that the gas soluble and gas permeable properties of the PDMS can be applied to design vacuum-driven power-free microfluidics [72]. It is likely that such properties could also be applied to extend plastron longevity.

To examine the plastron longevity on superhydrophobic PDMS surfaces, we performed an experimental study in which the sample was submerged in an undersaturated liquid. The plastron longevity was determined by measuring the percentage of surface area covered by gas. The effect of gas soluble property on the plastron longevity was examined by testing samples of different thicknesses, considering that a sample with an infinitely small thickness is similar to a non-gas soluble material. We will show that by increasing the sample thickness (i.e., increasing the total amount of gas dissolved in the PDMS sample), the plastron longevity increases. We will explain this trend by numerically solving the mass transfer between the PDMS surface and the liquid. Moreover, we will show that injecting gas through the PDMS extends the plastron longevity. Our work is novel since we demonstrate, for the first time, the effectiveness of using gas-soluble and gas-permeable materials, such as PDMS, to extend plastron longevity.

## 2. Materials and Methods

As shown in Figure 1, PDMS surfaces with two texture geometries (micro-holes and micro-posts) were used in this study. The micro-holes had a radius of *r* = 30 µm, a depth of *h* = 46 µm, and a wavelength of *λ* = 100 µm. The micro-posts had a radius of *r* = 30 µm, a height of *h* = 61 µm, and a wavelength of *λ* = 100 µm. The texture parameters are summarized in Table 1. In addition, Table 1 provides the surface energy difference between the Cassie–Baxter state and the Wenzel state per unit surface area and per surface tension, expressed as [73]:Δ*E =* (*ϕ_s_* − *r_W_*) cos*θ*_0_ − 1 + *ϕ_s_*,(1)
where *r_W_* is the Wenzel roughness, defined as the ratio of total surface area to projected surface area, *ϕ_s_* is the fraction of surface area covered by solid, and *θ*_0_ is the water contact angle on an ideally flat surface of the same material (here for PDMS, *θ*_0_ = 105° [74]). As shown in Table 1, for micro-holes, Δ*E* > 0, suggesting that the Cassie–Baxter state has a lower energy than the Wenzel state and is thus thermodynamically stable. In contrast, for micro-posts, Δ*E* < 0, indicating that the plastron is thermodynamically unstable. The water contact angles for PDMS surfaces with micro-holes and micro-posts, obtained by measuring a small water droplet resting on the samples, were 114° and 125°, respectively.

These PDMS surfaces were created using a standard soft-lithography procedure involving the following steps. First, the base and curing agents of PDMS (Dow SYLGARD 184, Dow, Midland, MI, USA) were mixed at a mass ratio of 10:1. Then, the mixture was degassed under vacuum for 10 min, gently poured onto an SU8 template, and crosslinked at 60 °C for 4 h. Finally, the PDMS surface was peeled off from the SU8 template. Since the PDMS is hydrophobic, no additional coatings were applied to modify the surface chemistry. As shown later, when submerged in water, gas was trapped within the surface texture, forming the plastron. According to the literature, the prepared PDMS has an oxygen solubility of 0.18 cm^3^/(cm^3^ atm) and a nitrogen solubility of 0.09 cm^3^/(cm^3^ atm) at standard temperature and pressure (STP) [72], and a nitrogen permeability of 1.34 × 10^−13^ mol/(Pa s m) [69]. Prior to the experimental tests, the PDMS samples were stored at atmospheric pressure for at least 2 days to reach equilibrium.

To examine the effect of the gas-soluble property of the PDMS surface on plastron longevity, we used an experimental setup illustrated in Figure 2a, which was also used in our previous work [14]. A PDMS surface (diameter 12.5 mm) was installed at the top of a tube filled with water. We varied the thickness of PDMS surface *d* from 0.9 to 6.0 mm: an infinitely thin sample can be approximated as a non-gas soluble surface, while a thicker sample stores more gas due to its larger volume and is expected to have greater plastron longevity. The textured side of the PDMS surface faced downward and was in contact with the water. The tube had a length of 270 mm, which was long enough for the gas molecules to diffuse freely in the direction perpendicular to the PDMS surface. Since the sample was located above the water surface and the hydrostatic pressure was close to *P_atm_*, the air concentration at the air–water interface was *c_i_* = *P_atm_*/*k_H_*, where *P_atm_* = 1 atm is the atmospheric pressure and *k_H_* is Henry’s law constant of air. To induce gas transfer from the plastron to the liquid, the air concentration in the ambient water, *c_∞_*, was set to be smaller than *c_i_*. The lower air concentration in the water was achieved by leaving a beaker of water under a vacuum for a certain duration and then pouring this degassed water into the tube.

Furthermore, to investigate whether the gas permeable property of the PDMS surface can be utilized to extend the plastron longevity, we performed experiments using the setup shown in Figure 2b. A PDMS surface (diameter 25.4 mm, thickness of *d* = 2.0 mm) was submerged in a tank filled with water. The tank had an inner dimension of 13 mm × 75 mm × 150 mm. The air concentration in the ambient water was set to be lower than that at the air–water interface (i.e., *c_∞_* < *c_i_*). To extend plastron longevity and sustain the plastron, gas was injected through the PDMS surface by connecting an air compressor to the back of the PDMS surface. The pressure at the back of the PDMS surface was varied between 1 and 3.1 atm and was measured by a high-precision pressure gauge (Omega Engineering Norwalk, CT, USA, #DPG108–030 G, range 30 psi, precision 0.25%). The pressure inside the tube was maintained close to 1 atm. Therefore, the pressure difference (*∆P*) across the two sides of the PDMS surface varied in the range of 0 < *∆P* < 2.1 atm.

In both setups shown in Figure 2, the air concentration in the ambient water was monitored throughout the experiment using an optical oxygen sensor (FirestingO2, Pyro Science, Aachen, Germany). To determine the plastron longevity, the status of plastrons on the PDMS surface was measured using a non-intrusive optical method. An LED light was used to illuminate the sample. The light beams reflected from the air–water interface and the PDMS–water interface were recorded by a CMOS camera (FLIR Grasshopper 3, pixel size 5.5 mm, 2048 × 2048 pixels). In the setup of Figure 2a, the field of view was 13 mm × 13 mm, just large enough to cover the entire PDMS surface. In the setup of Figure 2b, the field of view was 1.2 mm × 1.2 mm due to the use of a 10× objective lens. As will be shown later, the recorded image transitioned from bright to dark as the plastron slowly decayed. This was because the intensity of light reflected from the air–water interface was much larger than that reflected from the PDMS–water interface. The plastron longevity was determined based on the time when the percentage of surface area covered by gas fell below a certain value. All experiments were performed at room temperature of 20 ± 1 °C.

## 3. Results and Discussion

(a)Effect of PDMS Surface Thickness on Plastron Longevity

To examine the effect of the gas-soluble properties of the PDMS surface on plastron longevity, we performed experiments for PDMS surfaces with thicknesses ranging from 0.9 to 6 mm using the setup shown in Figure 2a. The gas concentration in water was fixed at *c_∞_* = 0.30*c_i_*. Figure 3a,b show the effect of *d* on the time variation in plastron status for PDMS with micro-holes and micro-posts, respectively. With increasing time (*t*), due to the transfer of gas from the plastron to the surrounding liquid, the surface area covered by gas (i.e., the bright regions in the image) decreased. Specifically, for PDMS with micro-holes, the gas was trapped within isolated micro-holes, creating numerous isolated plastrons. With increasing time, the number of bright dots (or the number of plastrons) decreased. For PDMS with micro-posts, the gas was trapped between the space of different posts, forming a single large plastron. With increasing time, the bright region (or the plastron) shrank in the horizontal direction along the surface. Furthermore, with increasing *d*, the wetting process was greatly delayed, indicating that a thicker PDMS surface had a longer plastron longevity.

To quantify the impact of *d* on the wetting processes, we processed the recorded images using a method from our previous work [14]. Briefly, the recorded images were binarized based on intensity, and the regions covered by gas were identified. We calculated the surface area covered by gas and defined it as *ϕ*_g_. We also defined *ϕ*_g0_ = *ϕ*_g_ (*t* = 0) as the surface area covered by gas at the beginning of the wetting process. Figure 4a,b show the effect of *d* on time variations in *ϕ*_g_/*ϕ*_g0_ for PDMS with micro-holes and micro-posts, respectively. Clearly, with increasing time, *ϕ*_g_ monotonically reduced to zero. However, the trends of *ϕ*_g_ for different textures were distinct: for micro-holes, *ϕ*_g_ decreased very slowly at first and then dropped dramatically; for micro-posts, *ϕ*_g_ decreased quickly at first, and the rate of decrease slowed over time. The possible reason for these difference trends at the beginning of the wetting transition is as follows: for micro-holes, the air–water interface mainly moved in the direction perpendicular to the surface (i.e., a hole fully filled with gas transitioned to a hole partially filled with gas), while for micro-posts, the air–water interface moved in the direction parallel to the surface (i.e., the plastron shrank in size in the horizontal direction). Although the two textures exhibited different trends in *ϕ*_g_, both showed a smaller rate of decay in *ϕ*_g_ as *d* increased.

To estimate the time scale of wetting processes, we defined the plastron longevity (*t_f_*) as the time when *ϕ*_g_/*ϕ*_g0_ = 0.05. The value of 0.05 was arbitrarily selected. The general conclusions of our work do not change if other values are used to define the plastron longevity. Figure 4c,d show *t_f_* as a function of *d* for PDMS with micro-holes and micro-posts, respectively. First, the PDMS surface with micro-posts had a longer plastron longevity than the PDMS surface with micro-holes. This was because the plastron volume on PDMS with micro-posts was larger than that on PDMS with micro-holes. A more detailed discussion of the dependence of plastron longevity on various texture parameters, including texture height, texture wavelength, and gas fraction, can be found in our previous work [15]. Second, regardless of the surface texture, the plastron longevity increased with increasing *d*. As *d* increased from 1 to 5 mm, *t_f_* increased by ~5 times from 160 s to 700 s for PDMS with micro-holes, and by ~2 times from 1740 s to 3890 s for PDMS with micro-posts. More interestingly, for PDMS with micro-holes, *t_f_* increased almost linearly with *d*. The effect of *d* on plastron longevity was larger for PDMS with micro-holes than for PDMS with micro-posts. This was probably because PDMS with micro-holes had a larger contact area between PDMS and liquid, enhancing the transfer of gas dissolved within the polymer chains to the undersaturated liquid. This contact area is *A_c_* = (1 − *πr*^2^/*λ*^2^)*A* = 0.71*A* and *A_c_* = (*πr*^2^/*λ*^2^)*A* = 0.29*A* for PDMS with micro-holes and micro-posts, respectively. Here, *A* is the cross-section area of the liquid, *r* is the radius of micro-holes and micro-posts, and *λ* is the texture wavelength.

To understand the mechanism behind the extension of plastron longevity due to the increased thickness of the PDMS surface, we performed a simplified numerical simulation, as shown in Figure 5a. We assumed a flat, smooth PDMS surface with thickness *d* exposed to liquid with a height of 10*d*. To match the experimental conditions, the initial air concentration in the PDMS surface was set to *c*_PDMS_ = 4.83 mol/m^3^, corresponding to a PDMS material saturated with air at atmospheric pressure and room temperature. The initial air concentration in water was set to *c_∞_* = 0.30*P_atm_*/*k_H_* = 0.238 mol/m^3^. Furthermore, the diffusion coefficients of air in PDMS and water were set to *D* = 3.4 × 10^−9^ m^2^/s [72] and 2.0 × 10^−9^ m^2^/s, respectively. Due to the difference between *c*_PDMS_ and *c_∞_*, air molecules diffused from the PDMS to the water, and the air concentration *c*(*t*, *y*) varied in both space and time. We defined *y* as the vertical coordinate, with *y* = 0 representing the PDMS–water interface. We numerically solved *c*(*t*, *y*) using COMSOL multi-physics simulations. Figure 5b,c show the concentration profiles at *t* = 200 s and the time evolution of mass flux at the PDMS–water interface, respectively. Results for *d* = 1 and 5 mm were shown. Clearly, for the case with a larger *d*, both the gas concentration and the mass flux near the PDMS–liquid interface were higher, suggesting that a thicker PDMS surface released more air into the water and reduced the degree of undersaturation near the plastron. These results explain why a thicker PDMS surface extends plastron longevity.

(b)Effect of Gas Injection Through PDMS on Plastron Longevity

To examine the effect of the gas permeable property of the PDMS surface on the plastron longevity, we performed experiments for PDMS surfaces using the setup shown in Figure 2b. The gas concentration in water was fixed at *c_∞_* = 0.30*c_i_*. The pressure difference across the two sides of the PDMS surface ranged from 0 to 2.1 atm. The sample thickness was constant *d* = 2.0 mm. The steady-state mass flux *J_S_* (mol/m^2^/s) through the PDMS due to the pressure and gas injection can be expressed by the following equation [69]:*J_S_* = *p ∆P/d*,(2)
where *∆P* is the pressure difference, *d* is the sample thickness, and *p* is the gas permeability of the PDMS. According to Equation (2), we expect that *J_S_* would increase linearly with increasing *∆P*. The plastron can be sustained or can grow when *J_S_* > *J_D_*, where *J_D_* denotes the mass flux of gas transferred from the plastron to the liquid due to diffusion. According to our previous work [14,15], during the wetting process, *J_D_* decreased over time following a power-law relation:*J_D_*(*t*)~*D*(*c_i_* − *c_∞_*)/*L_D_~t*^−0.5^,(3)
where *D* is the diffusion coefficient of gas in water, and *L_D_* is the diffusion length. The power-law exponent of *−*0.5 arises because the diffusion length increased with time as *L_D_*~(*Dt*)^0.5^, following a typical one-dimensional gas diffusion process.

Figure 6a,b show the time variations in plastron status on PDMS with micro-holes and micro-posts, respectively. For each texture, results for three different *∆P* values are shown. Clearly, with increasing *∆P*, the wetting process was significantly delayed, confirming that gas injection through the PDMS surface extended the plastron longevity. For micro-holes and *∆P =* 2.1 atm, the plastron grew with increasing time, suggesting that *J_S_* exceeded *J_D_*. Plastron longevity in this case was defined as the time when the plastron started to grow. As *∆P* increased, a steady state where the plastron remained in a constant shape was not observed. This is probably because, for any given *∆P*, *J_S_* remained constant while *J_D_* decreased over time, following Equation (3). For micro-post, although plastron longevity was extended due to gas injection, the plastron still decayed even at the highest-pressure case. The reason for this was likely that the plastron on micro-posts was not thermodynamically stable, as shown in Table 1.

To better quantify the effect of *∆P* on plastron longevity, we processed the images using the method discussed in the previous Section and calculated *ϕ*_g_. Figure 7a,b show the time variations in *ϕ*_g_/*ϕ*_g0_ for PDMS with micro-holes and micro-posts, respectively. For each texture, results for three different *∆P* values were shown. Clearly, with increasing *∆P*, it took a longer time for *ϕ*_g_ to decrease to zero, indicating a longer plastron longevity. Again, we defined *t_f_* as the time when *ϕ*_g_/*ϕ*_g0_ = 0.05. The general conclusions of our work do not change if other values are used to define the plastron longevity. As shown in Figure 7c,d, as *∆P* increased from 0 to 2.1 atm, *t_f_* increased by ~20 times from 35 s to 660 s for micro-holes, and by ~3 times from 1430 s to 4340 s for micro-posts. Furthermore, to confirm that at high pressure the mass flux due to gas injection through PDMS material was sufficiently large to extend plastron longevity (i.e., to confirm *J_S_* > *J_D_*), we estimated the time-averaged mass flux of gas transferred from the plastron to the liquid as:*J_D,ave_* = *m/t_f_A_g_*,(4)
where *m* is the total mass of gas trapped within one plastron, and *A_g_* is the surface area covered by gas (for micro-holes, *A_g_ = πr*^2^; for micro-posts, *A_g_ = λ*^2^ *− πr*^2^). Figure 7c,d show the variations in *J_S_*/*J_D_*_,*ave*_ as a function of *∆P* for micro-holes and micro-posts, respectively. As expected, with increasing *∆P*, *J_S_*/*J_D_*_,*ave*_ increased. Moreover, for large *∆P*, *J_S_*/*J_D_*_,*ave*_ > 1, confirming that the rate of gas replenishment was sufficiently large to counteract the plastron decay for PDMS with micro-holes.

(c)Effect of Gas Concentration in Liquid on Plastron Longevity

In the previous Sections, we observed that the plastron longevity was extended due to the gas-soluble and gas-permeable properties of PDMS material. All the experiments were performed under the condition where the gas concentration in water was fixed at *c_∞_* = 0.30*c_i_*. In this Section, we examine whether the observed trends hold true in water with different gas concentrations. To achieve this, we repeated the experiments using the setups shown in Figure 2 and used water with different gas concentrations. Figure 8a shows the plastron longevity as a function of *d* for *c_∞_* = 0.30*c_i_* and 0.45*c_i_*, and Figure 8b shows the plastron longevity as a function of *∆P* for *c_∞_* = 0.30*c_i_* and 0.60*c_i_*. All the data were collected for PDMS with micro-holes. The sample thickness for data shown in Figure 8b was 2 mm. As expected, for water with higher gas concentrations, the plastron longevity was longer. This is because increasing the gas concentration in water reduces the concentration difference near the gas–liquid interface and thus decreases the mass flux. Furthermore, regardless of the value of *c_∞_*, the trends that plastron longevity increases with increasing sample thickness and pressure difference still hold true. These results suggest that the gas soluble and gas permeable properties of PDMS material could be used to extend the plastron longevity in liquids with different gas concentrations.

## 4. Conclusions

In summary, we performed an experimental study on the longevity of plastrons on PDMS surfaces consisting of micro-holes and micro-posts when exposed to undersaturated liquid. The plastron longevity was determined by measuring the percentage of surface area covered by gas. First, we examined the effect of PDMS surface thickness on plastron longevity. We found that plastron longevity increased with increasing the sample thickness, suggesting that gas initially dissolved between the polymer chains of PDMS was transferred into the ambient liquid, delaying the wetting transition. Numerical simulations validated that a PDMS sample with a larger thickness released more gas from the polymer chains to the ambient liquid and resulted in a higher gas concentration near the PDMS–liquid interface. Second, we investigated the effect of gas injection through the PDMS material on plastron longevity. We found that plastron longevity increased with increasing the pressure difference across the PDMS sample, indicating that the gas permeable property of PDMS allowed the plastron to be replenished by gas injection. By calculating the mass flux caused by the gas injection through the PDMS material, we confirmed that it exceeded the mass flux caused by the diffusion of gas from the plastron to the surrounding liquid. In conclusion, our results demonstrate that plastron longevity can be extended by utilizing a gas-soluble and gas-permeable material. Our findings will have significant implications for various applications where maintaining a stable plastron is essential, such as the reduction in hydrodynamic friction drag by superhydrophobic surfaces.

Nevertheless, the current experiments were performed under a constant temperature, in a liquid with constant pressure, and with no flow. Given that plastron longevity is affected by many factors, including temperature, pressure, and flow speed, future studies are needed to validate the effectiveness of PDMS material for extending plastron longevity under different experimental conditions. For example, under flow conditions, the rate of gas transfer from plastron to undersaturated water (*J_D_*) is greatly accelerated due to mass convection, resulting in a shorter plastron longevity [28,29,51]. To maintain a stable plastron under flow conditions, a larger pressure difference and a smaller sample thickness may be required to satisfy the condition of *J_S_* = *J_D_*. Future studies should also investigate the long-term stability of PDMS material to assess its feasibility for practical applications.

## Figures and Tables

**Figure 1 biomimetics-10-00045-f001:**
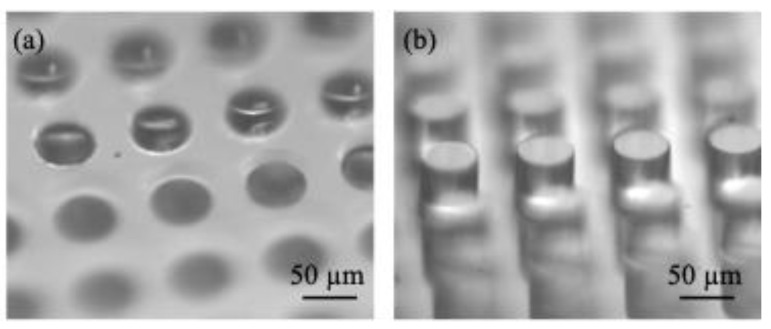
Microscope images of the surface texture on the PDMS surface with (**a**) micro-holes and (**b**) micro-posts.

**Figure 2 biomimetics-10-00045-f002:**
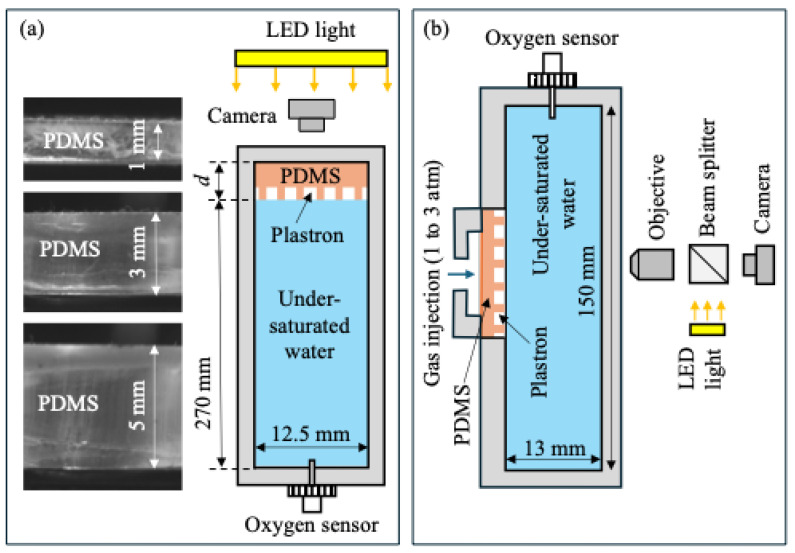
Experimental setups for (**a**) measuring the longevity of plastrons in undersaturated liquid; and (**b**) investigating the effect of gas injection on the longevity of plastrons in undersaturated liquid.

**Figure 3 biomimetics-10-00045-f003:**
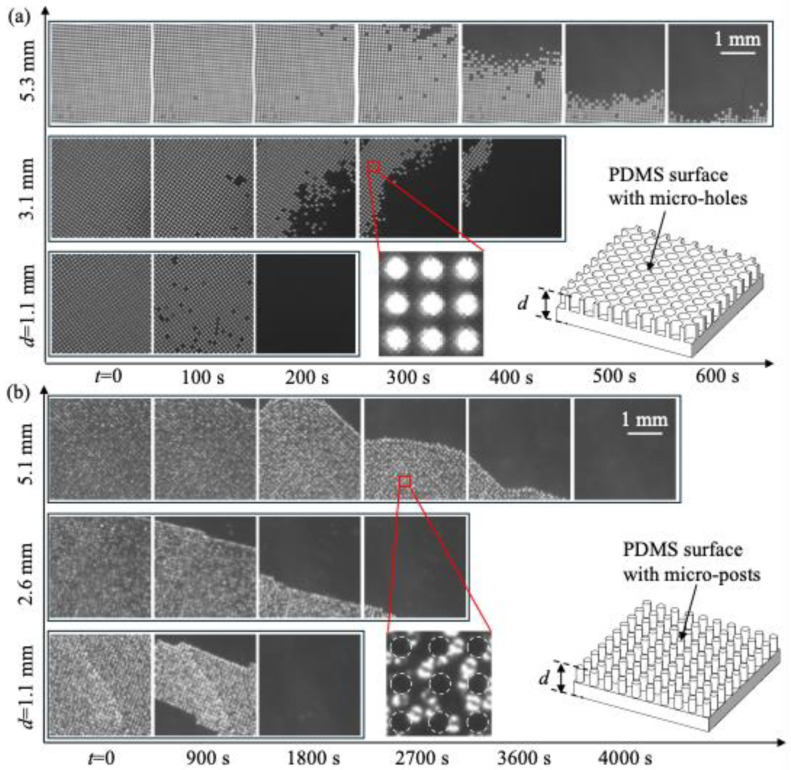
Effect of surface thickness (*d*) on the time-evolutions of plastron status for PDMS surface with micro-holes (**a**) and micro-posts (**b**) during the wetting process induced by gas diffusion. Results were obtained using the experimental setup shown in Figure 2a. Gas concentration in water was *c_∞_* = 0.30*c_i_*. The gas solubility and gas permeability were assumed to be independent of *d*.

**Figure 4 biomimetics-10-00045-f004:**
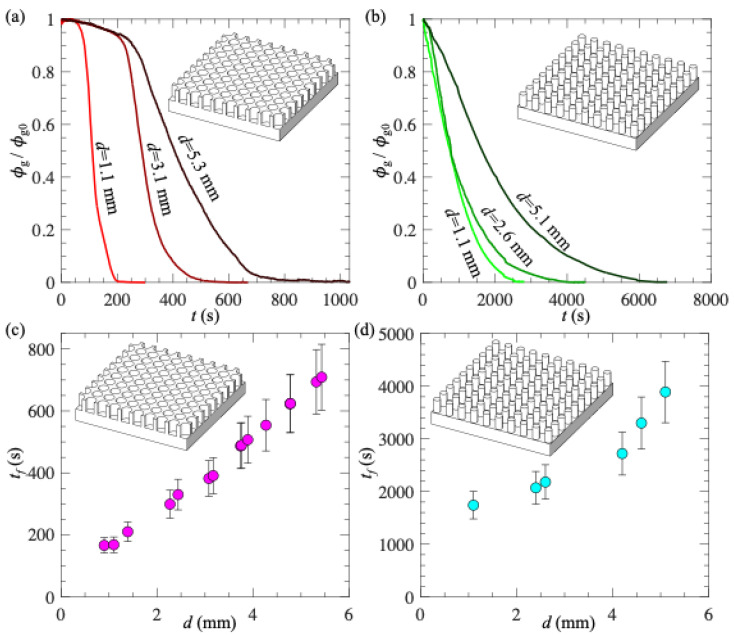
(**a**,**b**) Time evolutions of surface area coverage by gas (*ϕ*_g_/*ϕ*_g0_) for PDMS surface with micro-holes (**a**) and micro-posts (**b**) due to gas diffusion; (**c**,**d**) Plastron longevity as a function of surface thickness for PDMS surface with micro-holes (**c**) and micro-posts (**d**). Gas concentration in water was *c_∞_* = 0.30*c_i_*. In (**c**,**d**), each data point was obtained from 1 measurement, and the error bars were obtained by defining plastron longevity using *ϕ*_g_/*ϕ*_g0_ = 0.01 and 0.10.

**Figure 5 biomimetics-10-00045-f005:**
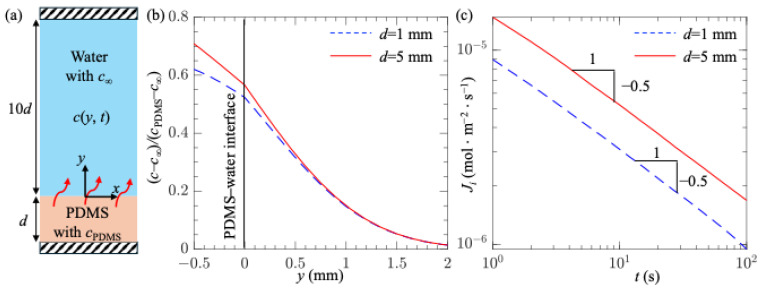
(**a**) A numerical model consisting of a PDMS surface with thickness *d* immersed in water with a height of 10*d*; (**b**) Gas concentration profiles at *t* = 200 s; and (**c**) Time variations in mass flux across the PDMS water interface *J_i_*. Results for *d* = 1 and 5 mm were shown.

**Figure 6 biomimetics-10-00045-f006:**
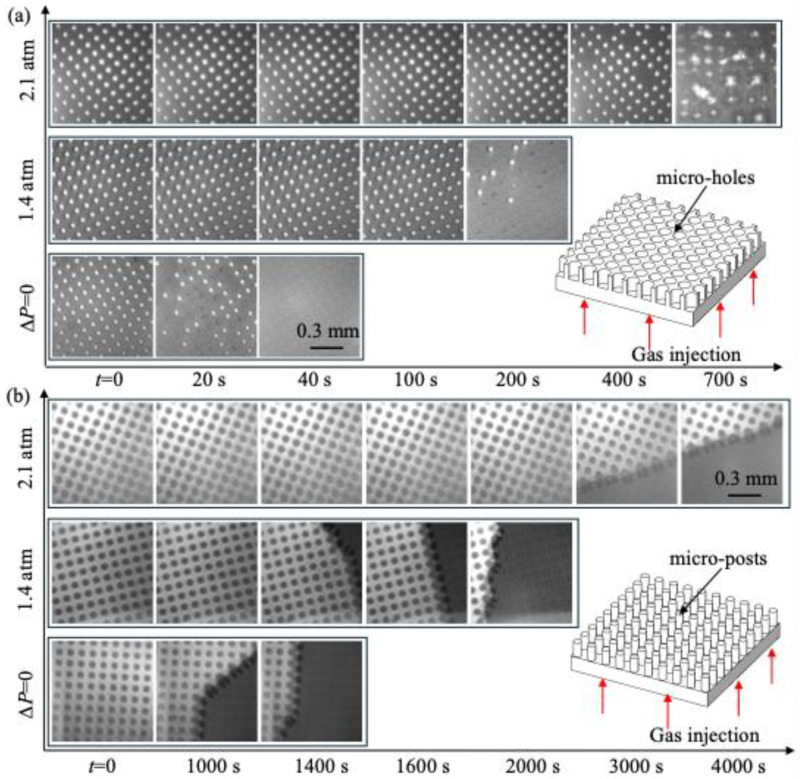
Effect of gas injection on the time-evolutions of plastron status for PDMS surface with micro-holes (**a**) and micro-posts (**b**) during the wetting process induced by gas diffusion. Results were obtained using the experimental setup shown in Figure 2b. The thickness of the PDMS surface was 2 mm, and the gas concentration in water was *c_∞_* = 0.30*c_i_*.

**Figure 7 biomimetics-10-00045-f007:**
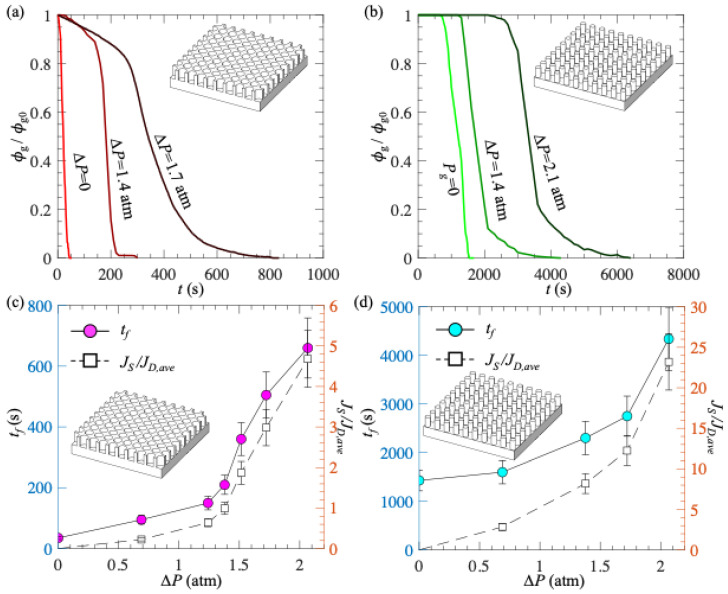
(**a**,**b**) Time evolutions of surface area coverage by gas (*ϕ*_g_/*ϕ*_g0_) for PDMS surface with micro-holes (**a**) and micro-posts (**b**) due to gas diffusion; (**c**,**d**) *t_f_* and *J_S_*/*J_D_*_,*ave*_ as a function of *∆P* for micro-holes (**c**) and micro-posts (**d**). The thickness of the PDMS surface was 2 mm, and the gas concentration in water was *c_∞_* = 0.30*c_i_*. In (**c**,**d**), each data point was obtained from 1 measurement, and the error bars were obtained by defining plastron longevity using *ϕ*_g_/*ϕ*_g0_ = 0.01 and 0.10.

**Figure 8 biomimetics-10-00045-f008:**
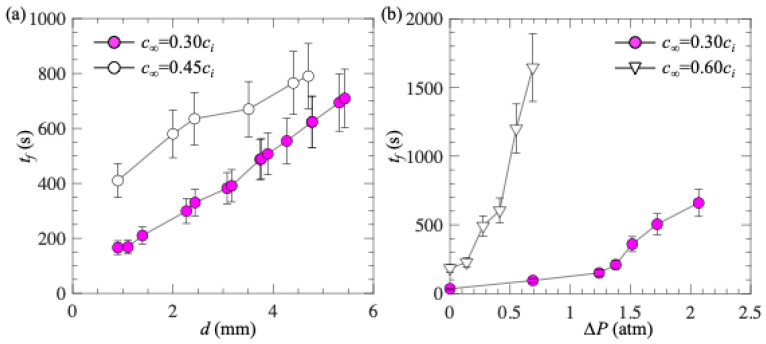
(**a**) Plastron longevity as a function of *d* for two different values of *c_∞_*; and (**b**) plastron longevity as a function of *∆P* for two different values of *c_∞_*. Results were obtained for PDMS with micro-holes. Sample thickness in (**b**) was 2 mm. Each data point corresponds to 1 measurement. The error bars were obtained by defining plastron longevity using *ϕ*_g_/*ϕ*_g0_ = 0.01 and 0.10.

**Table 1 biomimetics-10-00045-t001:** Texture parameters of micro-holes and micro-posts created on PDMS. Δ*E* > 0 means that the Wenzel state has a higher surface energy compared to the Cassie–Baxter state.

Samples	*r* (µm)	*h* (µm)	*λ* (µm)	Δ*E*
Micro-holes	30	46	100	0.015
Micro-posts	30	61	100	−0.45

## Data Availability

The original contributions presented in this study are included in the article. Further inquiries can be directed to the corresponding author.

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
