# Peer review of "Extend Plastron Longevity on Superhydrophobic Surface Using Gas Soluble and Gas Permeable Polydimethylsiloxane (PDMS)"

_biomimetics, 2025, doi:10.3390/biomimetics10010045_

Round 1

Reviewer 1 Report

Comments and Suggestions for Authors

In the paper entitled “Extended plastron longevity on the superhydrophobic surface using gas soluble and gas permeable PDMS”, the authors investigated and verified the effectiveness of PDMS on promoting the plastron longevity of underwater superhydrophobic surface both experimentally and numerically. The study is interesting, with practical significance in many fields including drag reduction, anti-biofouling, self-cleaning and so on. Thus, I recommend the publication of this manuscript in biomimetics after minor revision. And more specific questions are listed as follows:

1.      On page 5, the authors studied the effect of PDMS surface thickness on plastron longevity. However, several essential details are missing, such as the specific air concentration of the 'undersaturated water' under the experimental conditions and the variation in gas permeability of PDMS as a function of sample thickness. It is recommended that these details be included.

2.      How will the structural arrangement, such as the height, spacing, and diameter of the posts or holes affect the plastron longevity?

3.      The thickness of PDMS for Figure 6 should be provided in the caption.

4.      The error bars are missing for all the experimental results, which should be revised.

5.      How will the flow conditions of surrounding water affect the effectiveness of PDMS in promoting plastron longevity? Will it still work for flowing water?

Reviewer 2 Report

Comments and Suggestions for Authors

The manuscript is well-written, logically structured, and clearly presented. The experimental methods and results are described in detail, allowing readers to easily understand and replicate the experiments. The charts and graphs are clear and effectively support the arguments of the thesis. However, it would be advantageous to include additional insights on the future potential of the apparatus and methods discussed, particularly in the conclusion. Suggested improvements include:

  1. Diversity of Experimental Conditions: Conduct experiments under varying temperatures, pressures, or liquid environments to further validate the performance of PDMS materials in different conditions.
  2. Long-term Stability Study: While the paper primarily focuses on the short-term lifespan of the air film, it is recommended to investigate the long-term stability of PDMS materials to assess their feasibility for practical applications.

Lastly, I am curious about the water contact angle of the PDMS sample used by the authors. If measured, please provide the water contact angle value.

Reviewer 3 Report

Comments and Suggestions for Authors

1.The novelty of the work has not been specified.

2.There is no indication of the number of repetitions conducted, and error bars are not provided in the figures.

3.In page 7 line 206, to estimate the time scale of wetting processes, the authors defined the plastron longevity (tf) as the time when 𝜙g/𝜙g0=0.05. why choose this value?

4.Figure 3 and Figure 6 should be present with a better visibility.

5.In Figure 4 c and d, the tf values at d=3 mm was not given, please justify.

6.In line 216, the authors said that the different effect of d on plastron longevity between PDMS with micro-holes and with micro-posts was due to the different contact area between PDMS and liquid, could you please supply a theoretical calculation of an idea contact area of PDMS with micro-holes and with micro-posts based on the texture parameters to enhance this explanation?

Comments on the Quality of English Language

The English could be improved to more clearly express the research.

Round 2

Reviewer 3 Report

Comments and Suggestions for Authors

The required problems have been well addressed, I recommend  publicaiton without delay.